# Tan-Sun Transformation-Based Phase-Locked Loop in Detection of the Grid Synchronous Signals under Distorted Grid Conditions

**Guangjun Tan** †, **Chunan Zong** † and **Xiaofeng Sun** *,†

School of Electrical Engineering, Yanshan University, Qinhuangdao 066004, China; tan@ysu.edu.cn (G.T.); zongbonan@stumail.ysu.edu.cn (C.Z.)

* Correspondence: sxf@ysu.edu.cn; Tel.: +86-1309-137-0369

† These authors contributed equally to this work.

**Abstract:** When three-phase voltages are polluted with unbalance, DC offsets, or higher harmonics, it is a challenge to quickly detect their parameters such as phases, frequency, and amplitudes. This paper proposes a phase-locked loop (PLL) for the three-phase non-ideal voltages based on the decoupling network composed of two submodules. One submodule is used to detect the parameters of the fundamental and direct-current voltages based on Tan-Sun transformation, and the other is used to detect the parameters of the higher-harmonic voltages based on Clarke transformation. By selecting the proper decoupling vector by mapping Hilbert space to Euclidean space, the decoupling control for each estimated parameter can be realized. The settling time of the control law can be set the same for each estimated parameter to further improve the response speed of the whole PLL system. The system order equals the number of the estimated parameters in each submodule except that a low-pass filter is required to estimate the average amplitude of the fundamental voltages, so the whole PLL structure is very simple. The simulation and experimental results are provided in the end to validate the effectiveness of the proposed PLL technique in terms of the steady and transient performance.

**Keywords:** phase-locked loop; non-ideal voltages; Tan-Sun transformation; Hilbert space; Euclidean space; decoupling control; settling time

## 1. Introduction

With the widespread application of renewable energy and distributed generation (DG), more and more grid-connected inverters have been employed in the power grid. However, the power grid will be subject to some adverse impacts regarding the increased penetration level of DG and the excessive usage of single-phase, asymmetric, and non-linear loads, such as voltage unbalance, voltage distortion, frequency jumping, and so on [1,2]. In addition, DC offsets may appear in the detected grid voltage signals due to the saturation of voltage transducer, A/D conversion error, or power grid fault [3,4]. All the above problems will make it difficult for the grid-connected inverter to synchronize with the grid voltages normally, and the operation performance of the grid-connected inverter may deteriorate. Therefore, when the above problems occur in the grid voltages, it is an important guarantee to acquire the grid synchronization signals such as phases, frequency, and amplitudes of the grid voltages quickly and accurately for the reliable operation of the grid-connected inverter.

Among power grid synchronization techniques, phase-locked loop (PLL) is one of the most effective solutions. PLL is a control system which can generate the output signal synchronized with the phase of input signal. The most widely used PLL technique in the three-phase power electronic system is the synchronous reference frame PLL (SRF-PLL) [5,6]. SRF-PLL utilizes proportional-integral

(PI) controller as the loop filter, and it can quickly and accurately acquire the synchronization signals of the grid voltages when the three-phase grid voltages are balanced. However, the ability of the PI controller to suppress the interference component that may occur in the phase detector (PD) error is very limited. Therefore, when the grid voltages are unbalanced, or contain DC offsets or higher harmonics, second harmonic component, first harmonic component, and higher harmonic components will appear in the phase and frequency signals detected by SRF-PLL, respectively, which will reduce the operation performance of SRF-PLL [7–9].

In order to solve the above problems existing in SRF-PLL under non-ideal grid voltage conditions, some improved PLL techniques based on SRF-PLL have been proposed in the past few decades. In order to extract the positive-sequence (PS) q-axis component of the three-phase unbalanced voltage, and compensate it completely by using PI controller, some advanced PLL techniques have been proposed, such as decoupled double SRF-PLL (DDSRF-PLL) [10,11], double second-order generalized integrator PLL (DSOGI-PLL) [12–14], double complex-coefficient filter PLL (DCCF-PLL) [15,16], and three-phase enhanced PLL (3P-EPLL) [17–20]. The above-mentioned former three PLL techniques have been proved to be equivalent to each other through strict mathematical derivation, which are all six-order systems [21]. 3P-EPLL is composed of four single-phase EPLL (1P-EPLL) units, each of which is a third-order system, so the whole 3P-EPLL is up to a twelfth-order system.

Among the generalized integrator (GI)-based solutions, references [22–24] proposed DSOGI-PLL or DSOGI frequency-locked loop (FLL) to suppress the higher-harmonic voltages by adding a third integrator to reject the DC-offset voltages. Yada and Kumar [25] proposed dual SO-SOGI to eliminate the DC-offset voltages and reduce the higher-harmonic voltages. Zhang et al. [9] proposed mixed SOGI and third-order GI to eliminate the DC-offset voltages. Shah et al. [26] proposed fourth-order GI-FLL to reject the DC-offset voltages and filter the higher-harmonic voltages. However, the above GI-based PLL techniques can only reduce the higher-harmonic voltages, but cannot eliminate them completely. Moreover, the rejection speed of the DC-offset voltages and its impact on the operation performance of the whole GI-based PLL have not been referred to in the above solutions.

In order to completely eliminate the higher-harmonic voltages, the solutions based on decoupling network composed of multiple submodules have been proposed. Xiao et al. [27] proposed decoupled multiple SRFs to separate the PS, negative-sequence (NS), and higher-harmonic currents. Ali et al. [28] proposed a decoupled network composed of multiple SRFs to eliminate the NS and DC-offset voltages, and utilized the harmonic compensation network to eliminate the harmonic and interharmonic voltages. References [29,30] proposed a decoupling network composed of multiple SOGI to detect the PS and NS higher-harmonic voltages. References [15,31] proposed multiple CCF-PLL (MCCF-PLL) technique (or called as multiple adaptive vectorial filters FLL in [32]) to extract the PS, NS, and higher-harmonic components from the grid voltages. However, the above PLL techniques utilized a large number of submodules, and the order of each PLL technique is much higher than the number of the estimated parameters, so these PLL techniques are not easy to be implemented in the actual system.

In order to solve the problems existing in the above PLL techniques, a PLL technique based on the combination of Tan-Sun transformation [33,34] and Clarke transformation is proposed in this paper. The decoupling network constructed by the two submodules based on the above two transformations can be used to separate the fundamental components (FCs), DC-offset components (DCs), and higher-harmonic components (HCs) from the three-phase non-ideal grid voltages, respectively, and detect the synchronization signals of these components quickly and accurately. According to the corresponding relationship between the scalar in Hilbert space and the vector in Euclidean space, the PD errors of the above two submodules can be rewritten as the vector form. By selecting the vector which is perpendicular to the coefficient vector of the other estimated parameter errors in each PD vector error, the inner product of the selected vector and the PD vector error is related to only one of the estimated parameter errors, so the decoupling control

for each estimated parameter is realized. Both the estimated phase of phase A voltage, which is a ramp signal, and its differential signal estimated grid frequency need to be designed as second-order system; however, all the other estimated parameters only need to be designed as first-order systems. The system order equals the number of the estimated parameters in each submodule except that the estimation for the average amplitude of the fundamental voltages needs a low-pass filter (LPF), so the control structure is very simple. In addition, the response speed of the whole PLL system can be further improved by setting the control law with the same settling time for each estimated parameter. Finally, the simulation and experiments of the proposed PLL technique are carried out to verify the effectiveness of the proposed PLL technique in terms of the steady and transient performance.

## 2. The Whole PLL Structure with Decoupling Network of Two Submodules

In the three-phase three-wire power electronic system, the non-ideal grid voltages $e_{abc}$ can be expressed as the sum of FCs ($e_{abc1}$), DCs ($e_{abc0}$), and HCs ($e_{abch}$) as follows

$$e_a = \underbrace{\frac{2}{\sqrt{3}} E_m \sin(\varphi_{ba} - \varphi_{ca}) \cos\theta_a}_{e_{a1}} + \underbrace{E_{a0}}_{e_{a0}} + \underbrace{\sum_{\substack{n=6k\pm1}}^{k\in\mathbf{N}^+} E_{nm} \cos(n\theta_a + \varphi_n)}_{e_{ah}} \tag{1a}$$

$$e_b = \underbrace{\frac{2}{\sqrt{3}} E_m \sin\varphi_{ca} \cos(\theta_a + \varphi_{ba})}_{e_{b1}} + \underbrace{E_{b0}}_{e_{b0}} + \underbrace{\sum_{\substack{n=6k\pm1}}^{k\in\mathbf{N}^+} E_{nm} \cos\left[n\left(\theta_a - \frac{2\pi}{3}\right) + \varphi_n\right]}_{e_{bh}} \tag{1b}$$

$$e_c = \underbrace{-\frac{2}{\sqrt{3}} E_m \sin\varphi_{ba} \cos(\theta_a + \varphi_{ca})}_{e_{c1}} + \underbrace{(-E_{a0} - E_{b0})}_{e_{c0}} + \underbrace{\sum_{\substack{n=6k\pm1}}^{k\in\mathbf{N}^+} E_{nm} \cos\left[n\left(\theta_a + \frac{2\pi}{3}\right) + \varphi_n\right]}_{e_{ch}} \tag{1c}$$

In the first terms $e_{abc0}$ as shown in Equation (1), $E_m$ is the average amplitude of $e_{abc1}$, $\theta_a$ is the phase of $e_a$, and $\varphi_{ba}$ and $\varphi_{ca}$ are the phase differences between $e_b$ and $e_a$, and $e_c$ and $e_a$, respectively. As the three-phase three-wire system under study is free from zero-sequence (ZS) components, the amplitudes of $e_{abc1}$ are related to the phase differences $\varphi_{ba}$ and $\varphi_{ca}$, and they are $2/\sqrt{3}E_m \sin(\varphi_{ba} - \varphi_{ca})$, $2/\sqrt{3}E_m \sin\varphi_{ca}$, and $-2/\sqrt{3}E_m \sin\varphi_{ba}$, respectively. In the second terms $e_{abc0}$ as shown in Equation (1), the DC offsets of $e_a$ and $e_b$ can be set as $E_{a0}$ and $E_{b0}$, respectively, and the DC offset of $e_c$ can be set as $(-E_{a0} - E_{b0})$ as the system under study is free from ZS components. The third terms $e_{abch}$ as shown in Equation (1) are set to contain $[n = (6k \pm 1)]$-th ($k \in \mathbf{N}^+$) harmonic voltages, where $(6k + 1)$-th harmonics are set as PS, and $(6k - 1)$-th harmonics are set as NS. The amplitude of $n$-th harmonic is set as $E_{nm}$, while the initial phase of $n$-th harmonic is set as $\varphi_n$.

According to the expression form of the three-phase non-ideal voltages $e_{abc}$ as shown in Equation (1), this paper proposes two detection submodules (DSs), FC and DC detection submodule (FC&DC-DS) and HC detection submodule (HC-DS), to detect the above parameters contained in $e_{abc1}$, $e_{abc0}$, and $e_{abch}$, respectively. In these two DSs, FC&DC-DS is used to detect the parameters of $e_{abc1}$ and $e_{abc0}$ successively through Tan-Sun and Park transformation, and HC-DS is used to detect the parameters of $e_{abch}$ successively through Clarke and Park transformation. Then, the fundamental voltages with DC offsets and the higher-harmonic voltages can be reconstructed respectively according to the detected parameters in these two DSs, and the input voltages of each DS can be decoupled by cross-subtraction of the voltages reconstructed by the other DS. The whole PLL structure composed of FC&DC-DS and HC-DS with decoupling network is shown in Figure 1, and these two DSs will be designed respectively in the following sections.

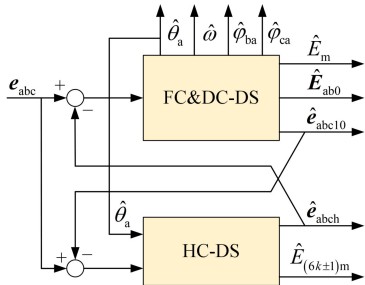

**Figure 1.** The whole phase-locked loop (PLL) structure.

## 3. FC-DS in Detection of Fundamental-Voltage Parameters

The PLL technique for detecting the parameters of $e_{abc1}$ in the FC&DC-DS will be briefly reviewed at first, whose detailed design process has been presented in [35]. $e_{abc1}$ can be rewritten according to Equation (1) as

$$e_{a1} = \frac{2}{\sqrt{3}} E_m \sin\left(\varphi_{ba} - \varphi_{ca}\right) \cos\theta_a \tag{2a}$$

$$e_{b1} = \frac{2}{\sqrt{3}} E_m \sin\varphi_{ca} \cos\left(\theta_a + \varphi_{ba}\right) \tag{2b}$$

$$e_{c1} = -\frac{2}{\sqrt{3}} E_m \sin\varphi_{ba} \cos\left(\theta_a + \varphi_{ca}\right) \tag{2c}$$

where, the actual parameter vector is denoted as $\boldsymbol{p}_1 = \begin{bmatrix} \theta_a & \varphi_{ba} & \varphi_{ca} \end{bmatrix}^{\mathrm{T}}$. The expression form of the PD error $e_{q1}$ can be calculated as the second element of the column matrix $\boldsymbol{e}_{dq1}$ derived successively though Tan-Sun and Park transformation of $\boldsymbol{e}_{abc1}$ as

$$\boldsymbol{e}_{dq1} = \boldsymbol{T}_{Park} \boldsymbol{T}_{Tan-Sun} \boldsymbol{e}_{abc1} \tag{3}$$

where

$$\boldsymbol{T}_{Tan-Sun} = \begin{bmatrix} \cos\left(-\hat{\varphi}_{ba} - \hat{\varphi}_{ca}\right) & \cos\left(\hat{\varphi}_{ba} - \hat{\varphi}_{ca}\right) & \cos\left(\hat{\varphi}_{ca} - \hat{\varphi}_{ba}\right) \\ \sin\left(-\hat{\varphi}_{ba} - \hat{\varphi}_{ca}\right) & \sin\left(\hat{\varphi}_{ba} - \hat{\varphi}_{ca}\right) & \sin\left(\hat{\varphi}_{ca} - \hat{\varphi}_{ba}\right) \end{bmatrix} \tag{4}$$

$$\boldsymbol{T}_{Park} = \begin{bmatrix} \cos\hat{\theta}_a & \sin\hat{\theta}_a \\ -\sin\hat{\theta}_a & \cos\hat{\theta}_a \end{bmatrix} \tag{5}$$

And the estimated parameter vector is denoted as $\hat{\boldsymbol{p}}_1 = \begin{bmatrix} \hat{\theta}_a & \hat{\varphi}_{ba} & \hat{\varphi}_{ca} \end{bmatrix}^{\mathrm{T}}$. Then, $e_{q1}$ can be approximated as the following linear function about the estimated parameter errors $\Delta\theta_a$, $\Delta\varphi_{ba}$, and $\Delta\varphi_{ca}$, the vector form of which is denoted as $\Delta\boldsymbol{p}_1 = \begin{bmatrix} \Delta\theta_a & \Delta\varphi_{ba} & \Delta\varphi_{ca} \end{bmatrix}^{\mathrm{T}}$, by Taylor expansion of it with respect to $\Delta\boldsymbol{p}_1$ about the point $\Delta\boldsymbol{p}_1 = \boldsymbol{0}$ as

$$
\begin{aligned}
e_{q1}\left(\Delta\boldsymbol{p_1}\right) \approx{}& \frac{E_m}{\sqrt{3}} \left[\sin 2\varphi_{ca} - \sin 2\varphi_{ba} + \sin 2\left(\varphi_{ba} - \varphi_{ca}\right)\right] \Delta\theta_a \\
&+ \frac{E_m}{\sqrt{3}} \left[\sin 2\varphi_{ca} - \sin 2\theta_a + \sin 2\left(\theta_a + \varphi_{ca}\right)\right] \Delta\varphi_{ba} \\
&+ \frac{E_m}{\sqrt{3}} \left[\sin 2\theta_a - \sin 2\varphi_{ba} - \sin 2\left(\theta_a + \varphi_{ba}\right)\right] \Delta\varphi_{ca}
\end{aligned}
\tag{6}
$$

As $e_{q1}$ in Equation (6) is the function about $\Delta p_1$ with more than one parameter, Hilbert space should be built here to design the FC-DS structure. In the real space $L^2[0, 2\pi]$, $f(x)$, $g(x) \in L^2[0, 2\pi]$. If the inner product formula is defined as

$$\langle f(x), g(x)\rangle = \frac{1}{2\pi} \int_0^{2\pi} f(x) g(x) \, \mathrm{d}x \tag{7}$$

then the function system

$$\left\{1, \sqrt{2}\cos x, \sqrt{2}\sin x, \sqrt{2}\cos 2x, \sqrt{2}\sin 2x, \cdots\right\} \tag{8}$$

is an orthonormal basis in $L^2[0, 2\pi]$ [36]. The inner product defined in Equation (7) denotes the average value of the product of $f(x)$ and $g(x)$ in a fundamental period. If only the first three elements in Equation (8) are considered, the orthonormal basis in Hilbert space constructed by these three elements has a corresponding relationship with the space rectangular coordinate system in Euclidean space, and these three elements are corresponding to three unit vectors $\vec{i}$, $\vec{j}$, and $\vec{k}$ of three-dimensional Euclidean space. Based on the above relationship, if we define $x = 2\theta_a$, $e_{q1}$ in Equation (6) can be expressed as the following vector form

$$\begin{aligned}
\vec{e}_{q1}(\Delta p_1) &\approx \frac{E_m}{\sqrt{3}}\left(\Delta\theta_a\vec{a} + \Delta\varphi_{ba}\vec{b} + \Delta\varphi_{ca}\vec{c}\right) \\
&= \frac{E_m}{\sqrt{3}}\left[\sin 2\varphi_{ca} - \sin 2\varphi_{ba} + \sin 2\left(\varphi_{ba} - \varphi_{ca}\right)\right]\vec{i}\Delta\theta_a \\
&\quad + \frac{E_m}{\sqrt{3}}\left[\sin 2\varphi_{ca}\vec{i} + \frac{1}{\sqrt{2}}\sin 2\varphi_{ca}\vec{j} + \frac{1}{\sqrt{2}}\left(\cos 2\varphi_{ca} - 1\right)\vec{k}\right]\Delta\varphi_{ba} \\
&\quad + \frac{E_m}{\sqrt{3}}\left[-\sin 2\varphi_{ba}\vec{i} - \frac{1}{\sqrt{2}}\sin 2\varphi_{ba}\vec{j} + \frac{1}{\sqrt{2}}\left(1 - \cos 2\varphi_{ba}\right)\vec{k}\right]\Delta\varphi_{ca}
\end{aligned} \tag{9}$$

The control laws for $\hat{\omega}$ and $\hat{p}_1$ can be expressed as the dot product of $\vec{e}_{q1}$ and certain vectors as

$$\frac{\mathrm{d}\hat{\omega}}{\mathrm{d}t} = -K_{1\omega}\vec{v}_1 \cdot \vec{e}_{q1} \tag{10a}$$

$$\frac{\mathrm{d}\hat{\theta}_a}{\mathrm{d}t} = \hat{\omega} - K_{1\theta}\vec{v}_1 \cdot \vec{e}_{q1} \tag{10b}$$

$$\frac{\mathrm{d}\hat{\varphi}_{ba}}{\mathrm{d}t} = -K_2\vec{v}_2 \cdot \vec{e}_{q1} \tag{10c}$$

$$\frac{\mathrm{d}\hat{\varphi}_{ca}}{\mathrm{d}t} = -K_3\vec{v}_3 \cdot \vec{e}_{q1} \tag{10d}$$

where, $K_{1\omega}$, $K_{1\theta}$, $K_2$, and $K_3$ are all greater than zero. In order to implement the decoupling control for each estimated parameter in the control laws Equation (10), $\vec{v}_1$ should be perpendicular to $\vec{b}$ and $\vec{c}$, $\vec{v}_2$ should be perpendicular to $\vec{c}$ and $\vec{a}$, and $\vec{v}_3$ should be perpendicular to $\vec{a}$ and $\vec{b}$, respectively. In order to simplify the design process, $\vec{v}_1$, $\vec{v}_2$, and $\vec{v}_3$ should be set as unit vectors, and the signs of these unit vectors should be set to ensure the stability of the control laws Equation (10). Based on the above analysis, $\vec{v}_1$, $\vec{v}_2$, and $\vec{v}_3$ can be calculated as

$$\vec{v}_1 = \frac{\left|\vec{b} \times \vec{c}\right|}{\vec{b} \times \vec{c}} = -\frac{\sqrt{3}}{3}\vec{i} + \frac{\sqrt{6}}{3}\vec{j} \tag{11a}$$

$$\vec{v}_2 = \frac{\left|\vec{c} \times \vec{a}\right|}{\vec{c} \times \vec{a}} = \vec{j}\sin\varphi_{ba} + \vec{k}\cos\varphi_{ba} \tag{11b}$$

$$\vec{v}_3 = \frac{\left|\vec{a} \times \vec{b}\right|}{\vec{a} \times \vec{b}} = -\vec{j}\sin\varphi_{ca} - \vec{k}\cos\varphi_{ca} \tag{11c}$$

By substituting Equation (11) into Equation (10), the control laws for $\hat{\boldsymbol{p}}_1$ can be rewritten as

$$\frac{d\hat{\omega}}{dt} = \frac{K_{1\omega}E_m}{3} \left[\sin 2\varphi_{ca} - \sin 2\varphi_{ba} + \sin 2\left(\varphi_{ba} - \varphi_{ca}\right)\right] \Delta\theta_a \tag{12a}$$

$$\frac{d\hat{\theta}_a}{dt} = \hat{\omega} + \frac{K_{1\theta}E_m}{3} \left[\sin 2\varphi_{ca} - \sin 2\varphi_{ba} + \sin 2\left(\varphi_{ba} - \varphi_{ca}\right)\right] \Delta\theta_a \tag{12b}$$

$$\frac{d\hat{\varphi}_{ba}}{dt} = \frac{K_2 E_m}{\sqrt{6}} \left[\cos \varphi_{ba} - \cos \left(\varphi_{ba} - 2\varphi_{ca}\right)\right] \Delta\varphi_{ba} \tag{12c}$$

$$\frac{d\hat{\varphi}_{ca}}{dt} = \frac{K_3 E_m}{\sqrt{6}} \left[\cos \varphi_{ca} - \cos \left(2\varphi_{ba} - \varphi_{ca}\right)\right] \Delta\varphi_{ca} \tag{12d}$$

The control laws for $\hat{\boldsymbol{p}}_1$ should have the same settling time $t_s$, and the damping ratio of second-order system is equal to its optimal value $\sqrt{2}/2$, so the gains $K_{1\omega}$, $K_{1\theta}$, $K_2$, and $K_3$ can be approximately selected in the case of three-phase balanced voltages ($\varphi_{ba} = -2\pi/3$, $\varphi_{ca} = 2\pi/3$, and $e_{d1} = 1.5E_m$) as

$$K_{1\omega} = \frac{67.0261}{E_m t_s^2 \left[\sin 2\varphi_{ba} - \sin 2\varphi_{ca} - \sin 2\left(\varphi_{ba} - \varphi_{ca}\right)\right]} \approx \frac{38.6975}{e_{d1} t_s^2} \tag{13a}$$

$$K_{1\theta} = \frac{20.0538}{E_m t_s \left[\sin 2\varphi_{ba} - \sin 2\varphi_{ca} - \sin 2\left(\varphi_{ba} - \varphi_{ca}\right)\right]} \approx \frac{11.5781}{e_{d1} t_s} \tag{13b}$$

$$K_2 = \frac{3\sqrt{6}}{E_m t_s \left[\cos \left(\varphi_{ba} - 2\varphi_{ca}\right) - \cos \varphi_{ba}\right]} \approx \frac{3\sqrt{6}}{e_{d1} t_s} \tag{13c}$$

$$K_3 = \frac{3\sqrt{6}}{E_m t_s \left[\cos \left(2\varphi_{ba} - \varphi_{ca}\right) - \cos \varphi_{ca}\right]} \approx \frac{3\sqrt{6}}{e_{d1} t_s} \tag{13d}$$

The control laws in Equation (10) can be rewritten as the following form by mapping the vectors $\vec{v}_1$, $\vec{v}_2$, and $\vec{v}_3$ in Euclidean space to their corresponding scalar functions of Equation (8) in Hilbert space

$$\frac{d\hat{\omega}}{dt} = \frac{22.342 e_{q1}}{e_{d1} t_s^2} \left(1 - 2\cos 2\theta_a\right) \tag{14a}$$

$$\frac{d\hat{\theta}_a}{dt} = \hat{\omega} + \frac{6.6846 e_{q1}}{e_{d1} t_s} \left(1 - 2\cos 2\theta_a\right) \tag{14b}$$

$$\frac{d\hat{\varphi}_{ba}}{dt} = -\frac{6\sqrt{3} e_{q1}}{e_{d1} t_s} \sin \left(2\theta_a + \varphi_{ba}\right) \tag{14c}$$

$$\frac{d\hat{\varphi}_{ca}}{dt} = \frac{6\sqrt{3} e_{q1}}{e_{d1} t_s} \sin \left(2\theta_a + \varphi_{ca}\right) \tag{14d}$$

The real parameters $\boldsymbol{p}_1$ in the control laws Equation (14) can all be replaced by their corresponding estimated parameters $\hat{\boldsymbol{p}}_1$, and the initial values of $\hat{\boldsymbol{p}}_1$ can be set as

$$\hat{\omega}(0) = 50 \times 2\pi \text{rad/s} \tag{15a}$$

$$\hat{\varphi}_{ba}(0) = -2\pi/3 \tag{15b}$$

$$\hat{\varphi}_{ca}(0) = 2\pi/3 \tag{15c}$$

According to the above design process, the FC-DS structure in the FC&DC-DS in detection of the fundamental-voltage parameters is shown in Figure 2.

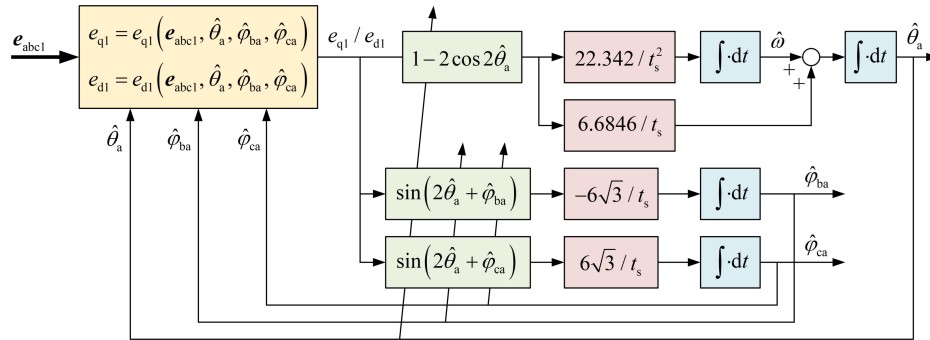

**Figure 2.** The fundamental components (FC)-detection submodules (DS) structure in detection of fundamental-voltage parameters.

## 4. FC&DC-DS in Detection of Fundamental Voltage and DC-Offset Voltage Parameters

Three-phase unbalanced voltages with only DC offsets, i.e., $e_{abc10}$, can be expressed according to Equation (1) as

$$e_{a10} = \frac{2}{\sqrt{3}} E_m \sin(\varphi_{ba} - \varphi_{ca}) \cos \theta_a + E_{a0} \tag{16a}$$

$$e_{b10} = \frac{2}{\sqrt{3}} E_m \sin \varphi_{ca} \cos(\theta_a + \varphi_{ba}) + E_{b0} \tag{16b}$$

$$e_{c10} = -\frac{2}{\sqrt{3}} E_m \sin \varphi_{ba} \cos(\theta_a + \varphi_{ca}) - E_{a0} - E_{b0}. \tag{16c}$$

In order to detect the DC offsets in three-phase voltages, the estimated values $\hat{E}_{a0}$ and $\hat{E}_{b0}$ of $E_{a0}$ and $E_{b0}$, respectively, can be constructed as

$$\hat{E}_{a0} = E_{a0} + \Delta E_{a0} \tag{17a}$$

$$\hat{E}_{b0} = E_{b0} + \Delta E_{b0} \tag{17b}$$

where, $\Delta E_{a0}$ and $\Delta E_{b0}$ are the estimated errors of $E_{a0}$ and $E_{b0}$, respectively. The fundamental components of $e_{abc10}$ in Equation (16) have been acquired by using FC-DS designed in Section 3, i.e.,

$$\hat{\theta}_a = \theta_a \tag{18a}$$

$$\hat{\varphi}_{ba} = \varphi_{ba} \tag{18b}$$

$$\hat{\varphi}_{ca} = \varphi_{ca} \tag{18c}$$

And only the control laws for $\hat{E}_{a0}$ and $\hat{E}_{b0}$ need to be designed in the FC&DC-DS. The actual parameter vector $\boldsymbol{p}_0$, the estimated parameter vector $\hat{\boldsymbol{p}}_0$, and the estimated parameter error vector $\Delta \boldsymbol{p}_0$ can be denoted respectively as $\boldsymbol{p}_0 = \begin{bmatrix} E_{a0} & E_{b0} \end{bmatrix}^T$, $\hat{\boldsymbol{p}}_0 = \begin{bmatrix} \hat{E}_{a0} & \hat{E}_{b0} \end{bmatrix}^T$, and $\Delta \boldsymbol{p}_0 = \begin{bmatrix} \Delta E_{a0} & \Delta E_{b0} \end{bmatrix}^T$. The expression form of the PD error $e_{q10}$ can be calculated as the second element of the column matrix $e_{dq10}$ derived successively though Tan-Sun and Park transformation of the differences between $e_{abc10}$ and $\hat{E}_{abc0}$ as

$$e_{dq10} = T_{Park} T_{Tan-Sun} (e_{abc10} - \hat{E}_{abc0}) \tag{19}$$

$e_{q10}$ in Equation (19) can be approximated as the following linear function about $\Delta \boldsymbol{p}_0$ by Taylor expansion of it with respect to $\Delta \boldsymbol{p}_0$ about the point $\Delta \boldsymbol{p}_0 = \boldsymbol{0}$ as

$$\begin{aligned} e_{q10}(\Delta \boldsymbol{p}_0) &\approx [\sin(\theta_a + \varphi_{ba} + \varphi_{ca}) - \sin(\theta_a + \varphi_{ba} - \varphi_{ca})] \Delta E_{a0} \\ &+ [\sin(\theta_a - \varphi_{ba} + \varphi_{ca}) - \sin(\theta_a + \varphi_{ba} - \varphi_{ca})] \Delta E_{b0} \end{aligned} \tag{20}$$

It can be derived according to Equation (20) that if $\Delta p_0 = 0$, then $e_{q10} = 0$; on the other hand, if $e_{q10} = 0$, then

$$\{[\sin(\varphi_{ba} + \varphi_{ca}) - \sin(\varphi_{ba} - \varphi_{ca})]\,\Delta E_{a0} - 2\sin(\varphi_{ba} - \varphi_{ca})\,\Delta E_{b0}\}\cos\theta_a$$
$$-2\sin\varphi_{ba}\sin\varphi_{ca}\Delta E_{a0}\sin\theta_a = 0 \qquad (21)$$

As $\cos\theta_a$ and $\sin\theta_a$ are time-variant while the coefficients of them in Equation (21) are time-invariant, Equation (21) can be equivalent to

$$A\Delta p_0 = 0 \qquad (22)$$

where

$$A = \begin{bmatrix} \sin(\varphi_{ba} + \varphi_{ca}) - \sin(\varphi_{ba} - \varphi_{ca}) & -2\sin(\varphi_{ba} - \varphi_{ca}) \\ -2\sin\varphi_{ba}\sin\varphi_{ca} & 0 \end{bmatrix} \qquad (23)$$

The determinant of the matrix $A$ can be calculated as

$$|A| = -4\sin\varphi_{ba}\sin\varphi_{ca}\sin(\varphi_{ba} - \varphi_{ca}) = 4\sin(\varphi_a - \varphi_b)\sin(\varphi_b - \varphi_c)\sin(\varphi_c - \varphi_a) \qquad (24)$$

In general, the difference between the instantaneous fundamental phases of any two voltages in $e_{abc10}$ is neither $0°$ nor $\pm180°$, so $|A|$ in Equation (24) is unequal to 0, and it can be derived according to Equation (22) that $\Delta p_0 = 0$. Through the above analysis, we can come to a conclusion that $\Delta p_0 = 0$ is equivalent to $e_{q10} = 0$. Therefore, if $e_{q10}$ is controlled to 0, the errors between $\hat{p}_0$ and $p_0$ are eliminated as well.

If the second and third elements in Equation (8) are considered, the orthonormal basis in Hilbert space constructed by these two elements has a corresponding relationship with the plane rectangular coordinate system in Euclidean space, and these two elements are corresponding to two unit vectors $\vec{l}$ and $\vec{m}$ of two-dimensional Euclidean space. Based on the above relationship, if we define $x = \theta_a$, $e_{q10}$ in Equation (20) can be expressed as the following vector form

$$\vec{e}_{q10}(\Delta p) \approx \Delta E_{a0}\vec{d} + \Delta E_{b0}\vec{e} = \sqrt{2}\sin\varphi_{ca}\left(\cos\varphi_{ba}\vec{l} - \sin\varphi_{ba}\vec{m}\right)\Delta E_{a0} - \sqrt{2}\sin(\varphi_{ba} - \varphi_{ca})\,\vec{l}\Delta E_{b0} \quad (25)$$

The control laws for $\hat{p}_0$ can be expressed as the dot products of $\vec{e}_{q10}$ and certain vectors as

$$\frac{d\hat{E}_{a0}}{dt} = -K_4\vec{v}_4 \cdot \vec{e}_{q10} \qquad (26a)$$

$$\frac{d\hat{E}_{b0}}{dt} = -K_5\vec{v}_5 \cdot \vec{e}_{q10} \qquad (26b)$$

where, $K_4$ and $K_5$ are both greater than zero. In order to implement the decoupling control for each estimated parameter in the control laws Equation (26), $\vec{v}_4$ should be perpendicular to $\vec{e}$, and $\vec{v}_5$ should be perpendicular to $\vec{d}$. In order to simplify the design process, $\vec{v}_4$ and $\vec{v}_5$ should be set as unit vectors, and the signs of these unit vectors should be set to ensure the stability of the control laws Equation (26). Based on the above analysis, $\vec{v}_4$ and $\vec{v}_5$ can be calculated as

$$\vec{v}_4 = \vec{m} \qquad (27a)$$

$$\vec{v}_5 = \sin\varphi_{ba}\vec{l} + \cos\varphi_{ba}\vec{m} \qquad (27b)$$

By substituting Equation (27) into Equation (26), the control laws for $\hat{p}_0$ can be rewritten as

$$\frac{d\hat{E}_{a0}}{dt} = \sqrt{2}K_4\sin\varphi_{ba}\sin\varphi_{ca}\Delta E_{a0} \qquad (28a)$$

$$\frac{d\hat{E}_{b0}}{dt} = \sqrt{2}K_5\sin\varphi_{ba}\sin(\varphi_{ba} - \varphi_{ca})\,\Delta E_{b0} \qquad (28b)$$

The control laws for $\hat{p}_0$ should have the same settling time $t_s$ as that of the control laws for $\hat{p}_1$, so the gains $K_4$ and $K_5$ can be approximately selected in the case of three-phase balanced voltages ($\varphi_{ba} = -2\pi/3$, $\varphi_{ca} = 2\pi/3$) as

$$K_4 = -\frac{3}{\sqrt{2}t_s \sin\varphi_{ba} \sin\varphi_{ca}} \approx \frac{2\sqrt{2}}{t_s} \tag{29a}$$

$$K_5 = -\frac{3}{\sqrt{2}t_s \sin\varphi_{ba} \sin(\varphi_{ba} - \varphi_{ca})} \approx \frac{2\sqrt{2}}{t_s} \tag{29b}$$

The control laws in Equation (26) can be rewritten as the following form by mapping the vectors $\vec{v}_4$ and $\vec{v}_5$ in Euclidean space to their corresponding scalar functions of Equation (8) in Hilbert space

$$\frac{d\hat{E}_{a0}}{dt} = -\sqrt{2}K_4 \sin\theta_a e_{q10} = -\frac{4e_{q10}}{t_s} \sin\theta_a \tag{30a}$$

$$\frac{d\hat{E}_{b0}}{dt} = -\sqrt{2}K_5 \sin(\theta_a + \varphi_{ba}) e_{q10} = -\frac{4e_{q10}}{t_s} \sin(\theta_a + \varphi_{ba}) \tag{30b}$$

The real parameters $p_1$ in the control laws Equation (30) can all be replaced by their corresponding estimated parameters $\hat{p}_1$.

According to the above design process, the FC&DC-DS in detection of both the fundamental-voltage parameters and the DC-offset voltage parameters is shown in Figure 3, and the reconstruction structure of the unbalanced voltages containing DC offsets is shown in Figure 4.

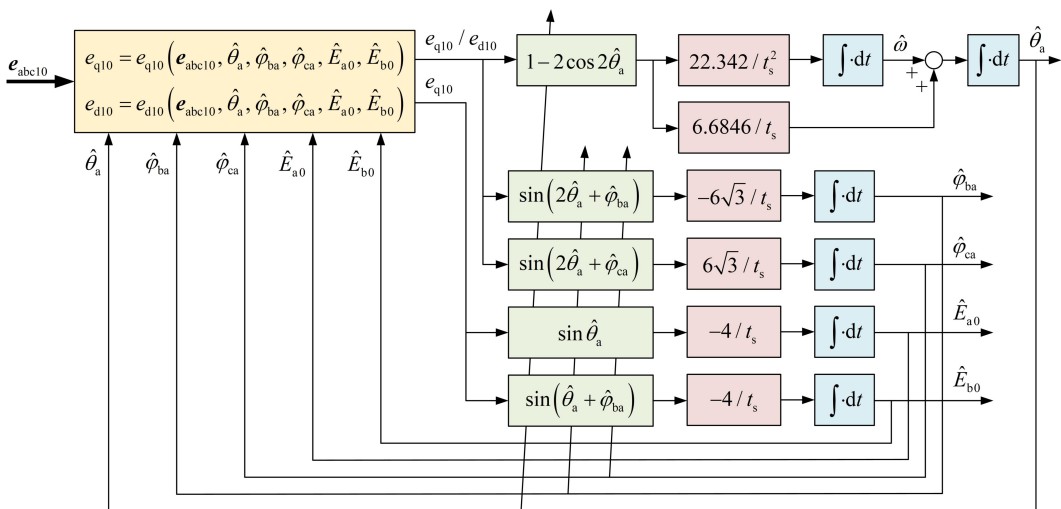

**Figure 3.** The FC&DC-DS in detection of the fundamental-voltage and DC-offset voltage parameters.

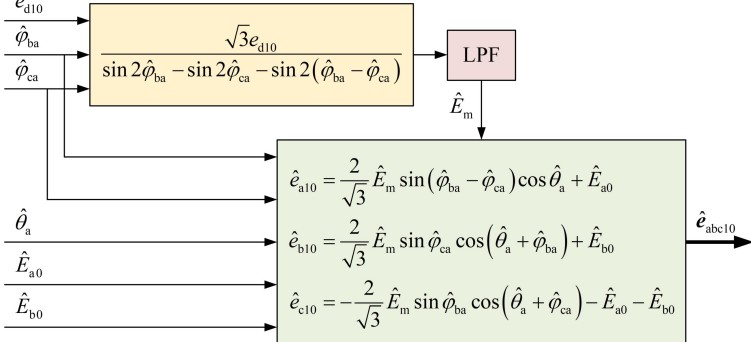

**Figure 4.** Reconstruction structure of the unbalanced voltages containing DC offsets.

In Figure 4, all the input variables are available through estimation except $e_{d10}$, which has a linear relationship with the average amplitude $E_m$ according to Equation (19). If $e_{d10}$ is directly feed forward to the reconstructed voltages of the FC&DC-DS, the whole PLL system cannot be stable due to the direct connection from the inputs to the outputs of the FC&DC-DS, which are then fed back to the inputs of the HC-DS. Therefore, a low-pass filter (LPF) should be added in the path for estimating $E_m$ to avoid this direct connection, the transfer function of which is shown as

$$G_{LPF} = \frac{s}{s + \omega_f} \tag{31}$$

where, $\omega_f$ is the bandwidth of the LPF, which should be set much higher than that of the control laws designed for all the above estimated parameters, so as to alleviate the impact of the LPF to the transient performance of the whole PLL system.

## 5. HC-DS in Detection of Higher-Harmonic Voltage Parameters

The higher harmonics in three-phase voltages can be expressed according to Equation (1) as

$$e_{ah} = \sum_{\substack{n=6k\pm1}}^{k\in\mathbf{N}^+} E_{nm} \cos\left(n\theta_a + \varphi_n\right) \tag{32a}$$

$$e_{bh} = \sum_{\substack{n=6k\pm1}}^{k\in\mathbf{N}^+} E_{nm} \cos\left[n\left(\theta_a - \frac{2\pi}{3}\right) + \varphi_n\right] \tag{32b}$$

$$e_{ch} = \sum_{\substack{n=6k\pm1}}^{k\in\mathbf{N}^+} E_{nm} \cos\left[n\left(\theta_a + \frac{2\pi}{3}\right) + \varphi_n\right] \tag{32c}$$

In order to facilitate the subsequent analysis, Equation (32) can be rewritten as the following form

$$e_{ah} = \sum_{\substack{n=6k\pm1}}^{k\in\mathbf{N}^+} \left(E_{ncm}\cos n\theta_a - E_{nsm}\sin n\theta_a\right) \tag{33a}$$

$$e_{bh} = \sum_{\substack{n=6k\pm1}}^{k\in\mathbf{N}^+} \left[E_{ncm}\cos n\left(\theta_a - \frac{2\pi}{3}\right) - E_{nsm}\sin n\left(\theta_a - \frac{2\pi}{3}\right)\right] \tag{33b}$$

$$e_{ch} = \sum_{\substack{n=6k\pm1}}^{k\in\mathbf{N}^+} \left[E_{ncm}\cos n\left(\theta_a + \frac{2\pi}{3}\right) - E_{nsm}\sin n\left(\theta_a + \frac{2\pi}{3}\right)\right] \tag{33c}$$

where

$$E_{ncm} = E_{nm}\cos\varphi_n \tag{34a}$$

$$E_{nsm} = E_{nm}\sin\varphi_n \tag{34b}$$

Clarke transformation matrix is

$$T_{Clarke} = \begin{bmatrix} \dfrac{2}{3} & -\dfrac{1}{3} & -\dfrac{1}{3} \\ 0 & \dfrac{\sqrt{3}}{3} & -\dfrac{\sqrt{3}}{3} \end{bmatrix} \tag{35}$$

According to Equation (5), Equation (33), and Equation (35), the dq-axes voltages $e_{dh}$ and $e_{qh}$ can be expressed as

$$
\begin{bmatrix} e_{dh} \\ e_{qh} \end{bmatrix} = \boldsymbol{T}_{Park}\boldsymbol{T}_{Clarke} \begin{bmatrix} e_{ah} \\ e_{bh} \\ e_{ch} \end{bmatrix}
$$

$$
= \begin{bmatrix} \sum\limits_{k\in\mathbf{N}^+} \left[ \left( E_{(6k+1)cm} + E_{(6k-1)cm} \right) \cos 6k\theta_a - \left( E_{(6k-1)sm} + E_{(6k+1)sm} \right) \sin 6k\theta_a \right] \\ \sum\limits_{k\in\mathbf{N}^+} \left[ \left( E_{(6k+1)sm} - E_{(6k-1)sm} \right) \cos 6k\theta_a - \left( E_{(6k-1)cm} - E_{(6k+1)cm} \right) \sin 6k\theta_a \right] \end{bmatrix} \tag{36}
$$

where, $\hat{\theta}_a$ in Park transformation matrix is provided by the FC&DC-DS, which has been acquired by using FC-DS designed in Section 3, i.e., $\hat{\theta}_a = \theta_a$. It can be seen from Equation (36) that the $(6k \pm 1)$-th harmonics in three-phase voltages Equation (32) are finally converted into $6k$-th harmonics in the dq-axes voltages Equation (36) successively through Clarke and Park transformation.

In order to simplify the expression form of $e_{dh}$ and $e_{qh}$ as shown in Equation (36), the relationships of the harmonic coefficients can be set as

$$
E_{d6kcm} = E_{(6k+1)cm} + E_{(6k-1)cm} \tag{37a}
$$

$$
E_{d6ksm} = E_{(6k-1)sm} + E_{(6k+1)sm} \tag{37b}
$$

$$
E_{q6kcm} = E_{(6k+1)sm} - E_{(6k-1)sm} \tag{37c}
$$

$$
E_{q6ksm} = E_{(6k-1)cm} - E_{(6k+1)cm} \tag{37d}
$$

By substituting Equation (37) into Equation (36), $e_{dh}$ and $e_{qh}$ can be rewritten as

$$
e_{dh} = \sum_{k\in\mathbf{N}^+} \left( E_{d6kcm} \cos 6k\theta_a - E_{d6ksm} \sin 6k\theta_a \right) \tag{38a}
$$

$$
e_{qh} = \sum_{k\in\mathbf{N}^+} \left( E_{q6kcm} \cos 6k\theta_a - E_{q6ksm} \sin 6k\theta_a \right) \tag{38b}
$$

The estimated values $\hat{E}_{d6kcm}$, $\hat{E}_{q6kcm}$, $\hat{E}_{d6ksm}$, $\hat{E}_{q6ksm}$ of the real parameters $E_{d6kcm}$, $E_{q6kcm}$, $E_{d6ksm}$, $E_{q6ksm}$, respectively, can be constructed as

$$
\hat{E}_{d6kcm} = E_{d6kcm} + \Delta E_{d6kcm} \tag{39a}
$$

$$
\hat{E}_{d6ksm} = E_{d6ksm} + \Delta E_{d6ksm} \tag{39b}
$$

$$
\hat{E}_{q6kcm} = E_{q6kcm} + \Delta E_{q6kcm} \tag{39c}
$$

$$
\hat{E}_{q6ksm} = E_{q6ksm} + \Delta E_{q6ksm} \tag{39d}
$$

where, $\Delta E_{d6kcm}$, $\Delta E_{q6kcm}$, $\Delta E_{d6ksm}$, $\Delta E_{q6ksm}$ are the estimated parameter errors of $E_{d6kcm}$, $E_{q6kcm}$, $E_{d6ksm}$, $E_{q6ksm}$, respectively. The real parameter vector $\boldsymbol{p}_h$, the estimated parameter vector $\hat{\boldsymbol{p}}_h$, and the estimated parameter error vector $\Delta\boldsymbol{p}_h$ can be denoted respectively as

$$
\boldsymbol{p}_h = \begin{bmatrix} [E_{d6kcm}]_{k\in\mathbf{N}^+} \\ [E_{d6ksm}]_{k\in\mathbf{N}^+} \\ [E_{q6kcm}]_{k\in\mathbf{N}^+} \\ [E_{q6ksm}]_{k\in\mathbf{N}^+} \end{bmatrix} \qquad \hat{\boldsymbol{p}}_h = \begin{bmatrix} [\hat{E}_{d6kcm}]_{k\in\mathbf{N}^+} \\ [\hat{E}_{d6ksm}]_{k\in\mathbf{N}^+} \\ [\hat{E}_{q6kcm}]_{k\in\mathbf{N}^+} \\ [\hat{E}_{q6ksm}]_{k\in\mathbf{N}^+} \end{bmatrix} \qquad \Delta\boldsymbol{p}_h = \begin{bmatrix} [\Delta E_{d6kcm}]_{k\in\mathbf{N}^+} \\ [\Delta E_{d6ksm}]_{k\in\mathbf{N}^+} \\ [\Delta E_{q6kcm}]_{k\in\mathbf{N}^+} \\ [\Delta E_{q6ksm}]_{k\in\mathbf{N}^+} \end{bmatrix}
$$

The PD errors $e_{dh}$ and $e_{qh}$ can be reconstructed according to Equation (38) as

$$
\begin{aligned}
e_{dh} &= \frac{2}{3}\left[e_{ah}\cos\theta_a + e_{bh}\cos\left(\theta_a - \frac{2\pi}{3}\right) + e_{ch}\cos\left(\theta_a + \frac{2\pi}{3}\right)\right] \\
&\quad - \sum_{k\in\mathbf{N}^+}\left(\hat{E}_{d6kcm}\cos 6k\theta_a - \hat{E}_{d6ksm}\sin 6k\theta_a\right) \\
&= \sum_{k\in\mathbf{N}^+}\left(\Delta E_{d6ksm}\sin 6k\theta_a - \Delta E_{d6kcm}\cos 6k\theta_a\right)
\end{aligned} \tag{40a}
$$

$$
\begin{aligned}
e_{qh} &= -\frac{2}{3}\left[e_{ah}\sin\theta_a + e_{bh}\sin\left(\theta_a - \frac{2\pi}{3}\right) + e_{ch}\sin\left(\theta_a + \frac{2\pi}{3}\right)\right] \\
&\quad - \sum_{k\in\mathbf{N}^+}\left(\hat{E}_{q6kcm}\cos 6k\theta_a - \hat{E}_{q6ksm}\sin 6k\theta_a\right) \\
&= \sum_{k\in\mathbf{N}^+}\left(\Delta E_{q6ksm}\sin 6k\theta_a - \Delta E_{q6kcm}\cos 6k\theta_a\right)
\end{aligned} \tag{40b}
$$

It can be derived according to Equation (40) that if $\Delta \boldsymbol{p}_h = \mathbf{0}$, then $e_{dh} = 0$, $e_{qh} = 0$; on the other hand, if $e_{dh} = 0$ and $e_{qh} = 0$, then

$$
\sum_{k\in\mathbf{N}^+}\left(\Delta E_{d6ksm}\sin 6k\theta_a - \Delta E_{d6kcm}\cos 6k\theta_a\right) = 0 \tag{41a}
$$

$$
\sum_{k\in\mathbf{N}^+}\left(\Delta E_{q6ksm}\sin 6k\theta_a - \Delta E_{q6kcm}\cos 6k\theta_a\right) = 0 \tag{41b}
$$

As $\cos 6k\theta_a$ and $\sin 6k\theta_a$ are time-variant while the coefficients of them in Equation (41) are time-invariant, it can be concluded that $\Delta E_{d6kcm}$, $\Delta E_{d6ksm}$, $\Delta E_{q6kcm}$, and $\Delta E_{q6ksm}$ all equal zero. Through the above analysis, we can come to a conclusion that $\Delta \boldsymbol{p}_h = \mathbf{0}$ is equivalent to $\Delta e_{dh} = 0$ and $\Delta e_{qh} = 0$. Therefore, if $\Delta e_{dh}$ and $\Delta e_{dh}$ are controlled to 0, the errors between $\hat{\boldsymbol{p}}_h$ and $\boldsymbol{p}_h$ are eliminated as well.

If we establish the relationship that $\cos 6k\theta_a$ and $\sin 6k\theta_a$ in Equation (8) are corresponding to $\vec{j}_k$ and $\vec{k}_k$ in Euclidean space, respectively, $e_{dh}$ and $e_{qh}$ in Equation (40) can be expressed as the following vector form

$$
\vec{e}_{dh}\left(\Delta \boldsymbol{p}\right) = \sum_{k\in\mathbf{N}^+}\left(\frac{1}{\sqrt{2}}\vec{k}_k\Delta E_{d6ksm} - \frac{1}{\sqrt{2}}\vec{j}_k\Delta E_{d6kcm}\right) \tag{42a}
$$

$$
\vec{e}_{qh}\left(\Delta \boldsymbol{p}\right) = \sum_{k\in\mathbf{N}^+}\left(\frac{1}{\sqrt{2}}\vec{k}_k\Delta E_{q6ksm} - \frac{1}{\sqrt{2}}\vec{j}_k\Delta E_{q6kcm}\right) \tag{42b}
$$

As the expression form of $\vec{e}_{dh}$ is the same as that of $\vec{e}_{qh}$, only the control laws for $\hat{E}_{d6kcm}$ and $\hat{E}_{d6ksm}$ need to be designed, and the control laws for $\hat{E}_{q6kcm}$ and $\hat{E}_{q6ksm}$ can be designed in the same way. The control laws for $\hat{E}_{d6kcm}$ and $\hat{E}_{d6ksm}$ can be expressed as the dot products of $\vec{e}_{dh}$ and certain vectors respectively as

$$
\frac{d\hat{E}_{d6kcm}}{dt} = -K_{dkc}\vec{v}_{dkc}\cdot\vec{e}_{dh} \tag{43a}
$$

$$
\frac{d\hat{E}_{d6ksm}}{dt} = -K_{dks}\vec{v}_{dks}\cdot\vec{e}_{dh} \tag{43b}
$$

where, $K_{dkc}$ and $K_{dks}$ are both greater than zero. For any $k\in\mathbf{N}^+$, if $\vec{v}_{dkc}$ and $\vec{v}_{dks}$ only contain the unit vectors $\vec{j}_k$ and $\vec{k}_k$, then $\vec{v}_{dkc}$ and $\vec{v}_{dks}$ will be certainly perpendicular to both $\{\vec{j}_n|n\in\mathbf{N}^+,\ n\neq k\}$ and $\{\vec{k}_n|n\in\mathbf{N}^+,\ n\neq k\}$. Therefore, in order to implement the decoupling control for $\hat{E}_{d6kcm}$ and $\hat{E}_{d6ksm}$ in the control laws Equation (43), $\vec{v}_{dkc}$ is only required to be perpendicular to $\vec{k}_k$, and $\vec{v}_{dks}$ is only required to be perpendicular to $\vec{j}_k$. In order to simplify the design process, $\vec{v}_{dkc}$ and $\vec{v}_{dks}$ should be set as unit

vectors, and the signs of these unit vectors should be set to ensure the stability of the control laws Equation (43). Based on the above analysis, $\vec{v}_{dkc}$ and $\vec{v}_{dks}$ can be calculated as

$$\vec{v}_{dkc} = -\vec{j}_k \tag{44a}$$

$$\vec{v}_{dks} = \vec{k}_k \tag{44b}$$

By substituting Equation (44) into Equation (43), the control laws for $\hat{E}_{d6kcm}$ and $\hat{E}_{d6ksm}$ can be rewritten as

$$\frac{d\hat{E}_{d6kcm}}{dt} = -\frac{1}{\sqrt{2}} K_{dkc} \Delta E_{d6kcm} \tag{45a}$$

$$\frac{d\hat{E}_{d6ksm}}{dt} = -\frac{1}{\sqrt{2}} K_{dks} \Delta E_{d6ksm} \tag{45b}$$

The control laws for $\hat{E}_{d6kcm}$ and $\hat{E}_{d6ksm}$ should have the same settling time $t_s$ as that of the control laws for $\hat{p}_1$ and $\hat{p}_0$, so the gains $K_{dkc}$ and $K_{dks}$ can be selected as

$$K_{dkc} = K_{dks} = \frac{3\sqrt{2}}{t_s} \tag{46}$$

The control laws in Equation (43) can be rewritten as the following form by mapping the vectors $\vec{v}_{dkc}$ and $\vec{v}_{dks}$ in Euclidean space to their corresponding scalar functions of Equation (8) in Hilbert space

$$\frac{d\hat{E}_{d6kcm}}{dt} = \sqrt{2} K_{dkc} e_{dh} \cos 6k\theta_a = \frac{6}{t_s} e_{dh} \cos 6k\theta_a \tag{47a}$$

$$\frac{d\hat{E}_{d6ksm}}{dt} = -\sqrt{2} K_{dks} e_{dh} \sin 6k\theta_a = -\frac{6}{t_s} e_{dh} \sin 6k\theta_a \tag{47b}$$

The control laws for $\hat{E}_{q6kcm}$ and $\hat{E}_{q6ksm}$ can be derived in the same way as the ones for $\hat{E}_{d6kcm}$ and $\hat{E}_{d6ksm}$ as

$$\frac{d\hat{E}_{q6kcm}}{dt} = \frac{6}{t_s} e_{qh} \cos 6k\theta_a \tag{48a}$$

$$\frac{d\hat{E}_{q6ksm}}{dt} = -\frac{6}{t_s} e_{qh} \sin 6k\theta_a \tag{48b}$$

According to the above design process, the HC-DS in detection of the higher-harmonic voltage parameters is shown in Figure 5, and the reconstruction structure of the higher-harmonic voltages is shown in Figure 6, where the sine and cosine coefficients of $(6k \pm 1)$-th harmonic voltages can be calculated according to Equation (37) as shown in Figure 6.

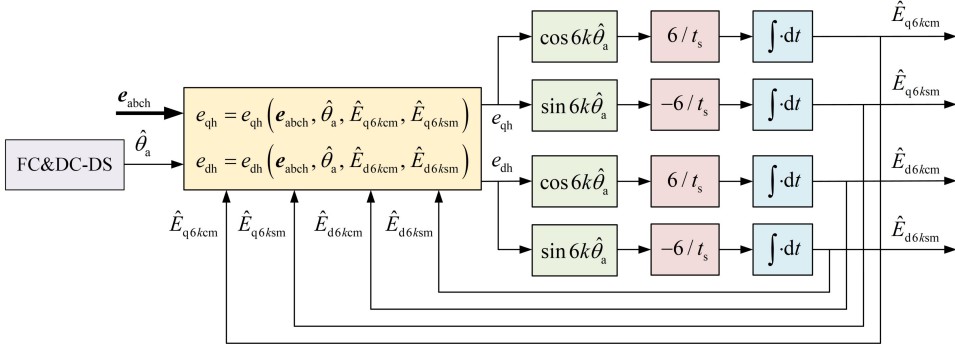

**Figure 5.** The higher-harmonic components (HC)-DS in detection of the higher-harmonic voltage parameters.

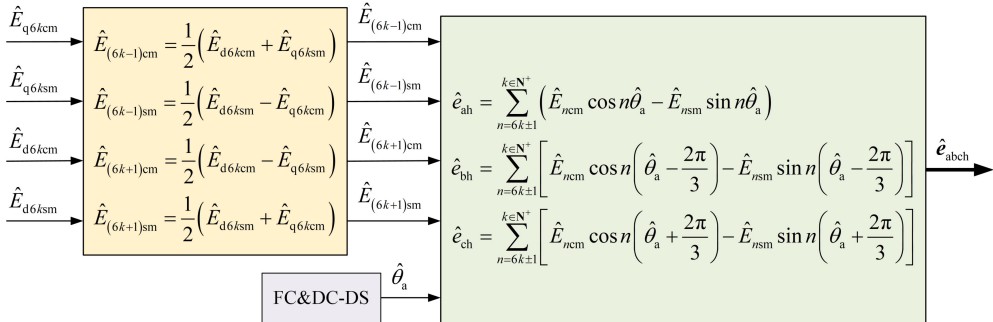

**Figure 6.** Reconstruction structure of the higher-harmonic voltages.

## 6. Simulation Analysis

This section will validate the correctness and effectiveness of the proposed whole PLL system, which is based on the combination of FC&DC-DS and HC-DS with decoupling network in MATLAB/Simulink. In the simulation, three-phase voltages are set unbalanced with DC offsets and higher harmonics, and the parameters of the three-phase non-ideal voltages are set as shown in Table 1. In Table 1, the fundamental voltage unbalance factor can be calculated as 14.58% according to [37]. In order to simulate the actual three-phase distorted grid voltages, the three-phase voltages mainly contain the 5th, 7th, 11th, and 13th harmonics, and the 5th and 11th harmonic voltages are NS ($-5$ and $-11$) while the 7th and 13th harmonic voltages are PS ($+7$ and $+13$). The amplitude of each order higher-harmonic voltage decreases as the harmonic order increase. The amplitudes of the 5th and 7th harmonic voltages, as well as those of the 11th and 13th harmonic voltages, are both nearly the same. The selection of the DC offsets in the three-phase voltages can refer to [3,16].

**Table 1.** Simulation Parameters.

| Symbol | Description | Value |
|---|---|---|
| $E_m$ | Average voltage amplitude | 100 V |
| $f$ | Grid frequency | 50 Hz |
| $\varphi_{ba}$ | Phase difference between phase B and A voltages | $-130°$ |
| $\varphi_{ca}$ | Phase difference between phase C and A voltages | $125°$ |
| $E_{a0}$ | DC offset in phase A voltage | 5 V |
| $E_{b0}$ | DC offset in phase B voltage | $-2$ V |
| $E_{5m}$ | Amplitude of 5th harmonic voltage | 12 V |
| $E_{7m}$ | Amplitude of 7th harmonic voltage | 11.5 V |
| $E_{11m}$ | Amplitude of 11th harmonic voltage | 4 V |
| $E_{13m}$ | Amplitude of 13th harmonic voltage | 3.8 V |

In the simulation, the settling time $t_s$ of the control law for each estimated parameter is set as 35 ms, and the bandwidth $\omega_f$ of the LPF for estimating $\hat{E}_m$ is set as 2000 rad/s. The simulation for the

proposed whole PLL system is performed under the following three different transient conditions of the three-phase non-ideal voltages, so as to validate the steady performance and response speed of the whole PLL system, respectively.

### 6.1. Three-Phase Unbalanced Voltages Being Injected with DC Offsets and Higher Harmonics

At $t = 0.3$ s, the three-phase unbalanced voltages are suddenly injected with DC offsets and higher harmonics, and the simulation results are shown in Figure 7. It can be seen from Figure 7 that the detection time for each parameter is less than 50 ms. The estimated error of each fundamental-voltage phase is almost zero, and the maximum estimated error of the grid frequency is less than 0.9 Hz.

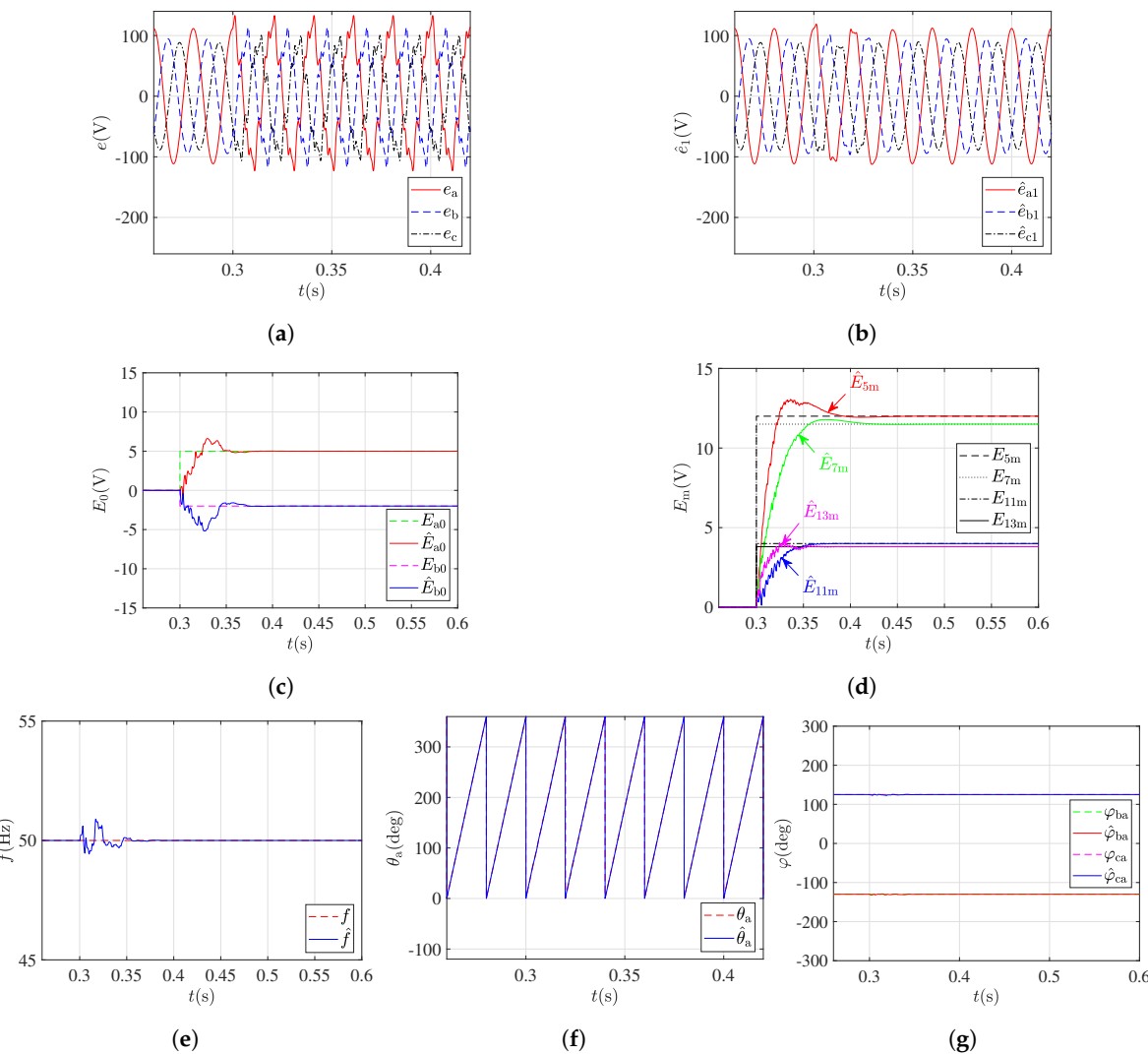

**Figure 7.** Simulation results when three-phase unbalanced voltages are suddenly injected with DC offsets and higher harmonics. The waveforms of (a) $e_{abc}$; (b) $\hat{e}_{abc1}$. The actual and estimated parameters of (c) $E_{a0}$, $E_{b0}$; (d) $E_{5m}$, $E_{7m}$, $E_{11m}$, $E_{13m}$; (e) $f$; (f) $\theta_a$; (g) $\varphi_{ba}$, $\varphi_{ca}$.

### 6.2. The Phase Difference between Phase B and Phase A Voltages Having a Step Change

At $t = 0.8$ s, the phase difference between phase B and phase A voltages has a step change from $-130°$ to $-150°$, and the simulation results are shown in Figure 8. It can be seen from Figure 8 that the detection time for each harmonic amplitude is about 60 ms, while the detection error is little, and the detection time for the other parameters is about 30 ms. The estimated error of each fundamental-voltage phase is very little, and the maximum estimated error of the grid frequency is less than 1.2 Hz.

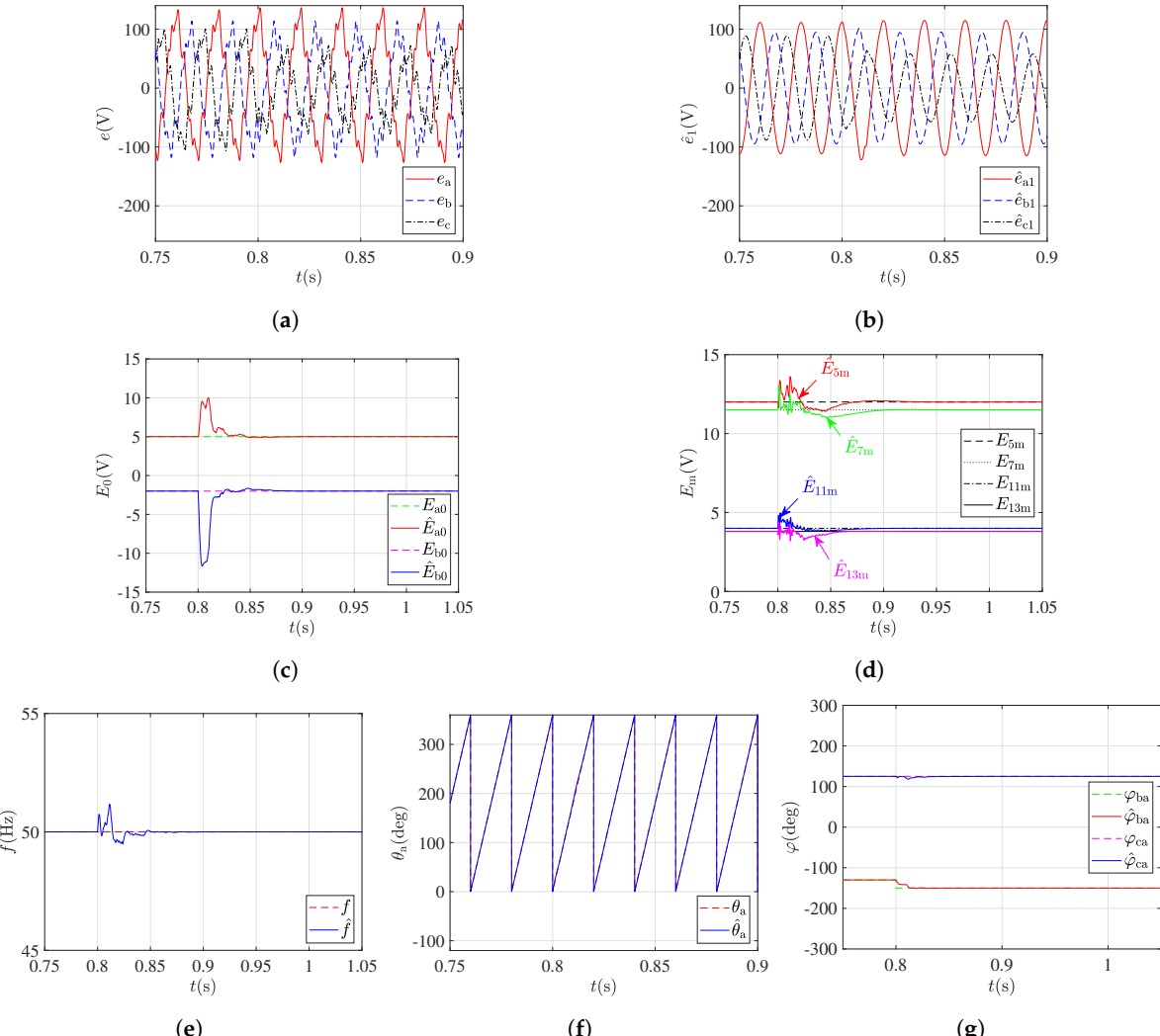

**Figure 8.** Simulation results when $\varphi_{ba}$ has a step change from $-130°$ to $-150°$. The waveforms of (**a**) $e_{abc}$; (**b**) $\hat{e}_{abc1}$. The actual and estimated parameters of (**c**) $E_{a0}$, $E_{b0}$; (**d**) $E_{5m}$, $E_{7m}$, $E_{11m}$, $E_{13m}$; (**e**) $f$; (**f**) $\theta_a$; (**g**) $\varphi_{ba}$, $\varphi_{ca}$.

### 6.3. The Grid Frequency Having a Step Change

At $t = 0.8$ s, the grid frequency $f$ has a step change from 50 Hz to 45 Hz, and the simulation results are shown in Figure 9. It can be seen from Figure 9 that the detection time for each parameter is about 60 ms. The estimated error of each fundamental-voltage phase is very little, and the maximum estimated error of the grid frequency is less than 1 Hz.

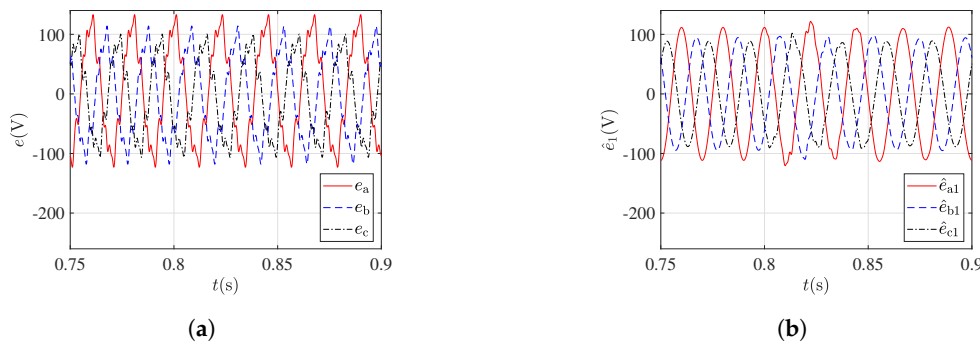

**Figure 9.** *Cont.*

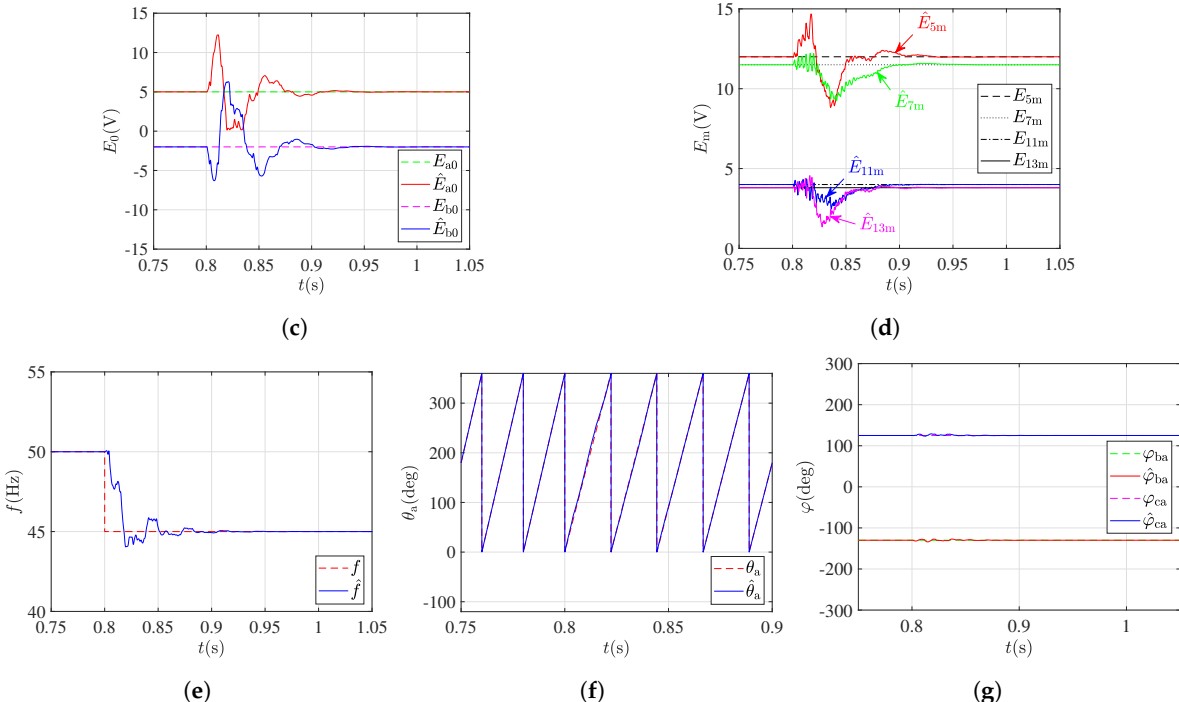

**Figure 9.** Simulation results when the grid frequency $f$ has a step change from 50 Hz to 45 Hz. The waveforms of (**a**) $e_{abc}$; (**b**) $\hat{e}_{abc1}$. The actual and estimated parameters of (**c**) $E_{a0}$, $E_{b0}$; (**d**) $E_{5m}$, $E_{7m}$, $E_{11m}$, $E_{13m}$; (**e**) $f$; (**f**) $\theta_a$; (**g**) $\varphi_{ba}$, $\varphi_{ca}$.

It can be concluded according to the simulation results that the proposed whole PLL system can successfully separate the FCs, DCs, and HCs of the three-phase non-ideal voltages, respectively, and detect all the parameters in less than 60ms under various types of transient conditions. Except for the big overshoots in detecting the DC offsets and higher harmonics of the three-phase voltages, the overshoots in detecting the other parameters are relatively very little. It is worth to note from the simulation results that the proposed whole PLL system has good steady and transient performance.

## 7. Experimental Results

The performance of the proposed whole PLL system has been evaluated through simulation in Section 6. This section will validate the correctness and effectiveness of the whole PLL system on the digital experimental platform. The experimental platform, as shown in Figure 10, is composed of a waveform generation board and a waveform detection board which both take the digital signal controller TMS320F28335 as the core. The waveform generation board is used to simulate the three-phase voltages sampled from the voltage transducers, and the parameters of the three-phase voltages are set as the same as those used in the simulation (Table 1). The waveform detection board is used to receive the three-phase voltages produced by the waveform generation board and detect the parameters of the received voltages. The settling time of each control law and the bandwidth of the LPF programmed in the waveform detection board are set also the same as those used in the simulation. This section will first validate the steady performance of the whole PLL system when both the DC offsets and higher harmonics are contained in the three-phase unbalanced voltages. And then, the transient performance of the whole PLL system will be validated under three different transient conditions, which are set as the same conditions as in the simulation.

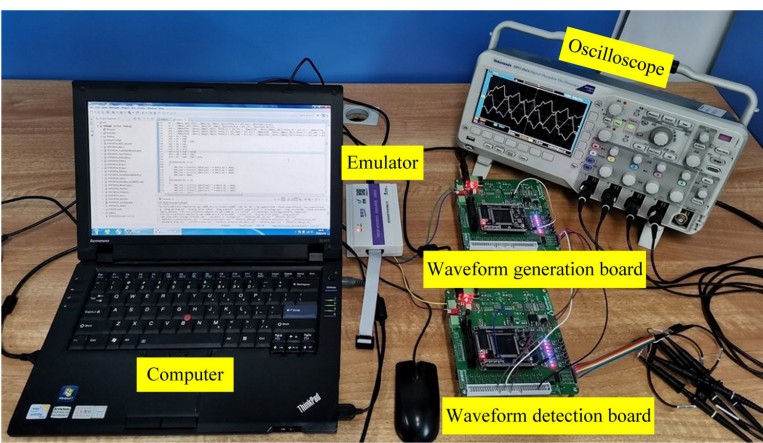

**Figure 10.** Photo of digital experimental platform.

*7.1. Analysis of the Steady Experimental Results*

When the three-phase unbalanced voltages contain DC offsets and higher harmonics, the steady experimental results are shown in Figure 11. It can be seen from Figure 11 that the whole PLL system can successfully separate the FCs, DCs, and HCs of the three-phase non-ideal voltages, respectively, and detect all the parameters without steady errors, which can demonstrate that the whole PLL system has good steady performance.

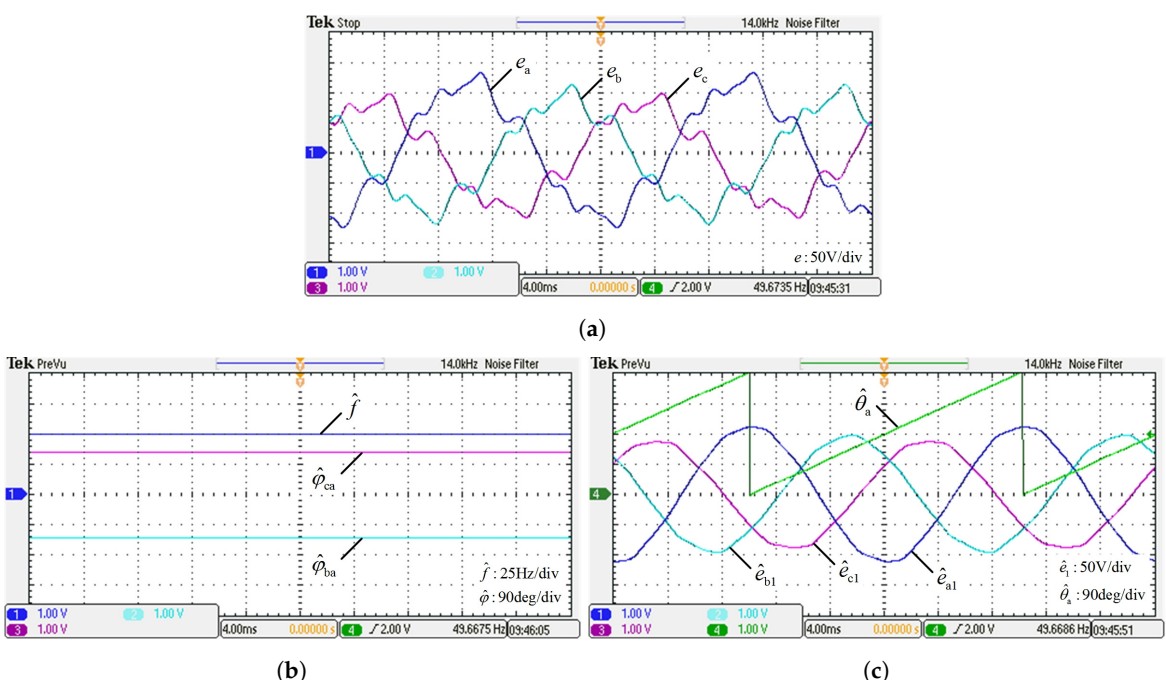

**Figure 11.** *Cont.*

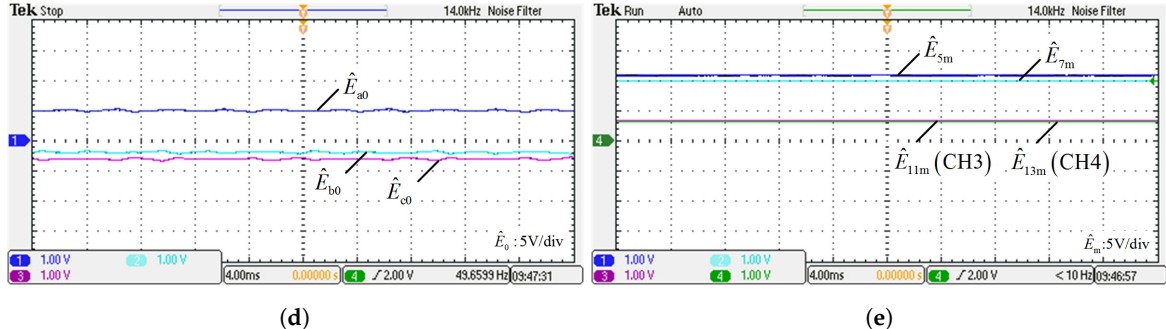

(**d**)             (**e**)

**Figure 11.** Experimental steady results when three-phase unbalanced voltages contain DC offsets and higher harmonics. The waveforms of (**a**) $e_{abc}$; (**b**) $\hat{f}$, $\hat{\varphi}_{ba}$, and $\hat{\varphi}_{ca}$; (**c**) $\hat{e}_{abc}$ and $\hat{\theta}_a$; (**d**) $\hat{E}_{abc0}$; (**e**) $\hat{E}_{5m}$, $\hat{E}_{7m}$, $\hat{E}_{11m}$, and $\hat{E}_{13m}$.

## 7.2. Analysis of the Transient Experimental Results

### 7.2.1. Condition I

When the three-phase unbalanced voltages are suddenly injected with DC offsets and higher harmonics, the experimental results are shown in Figure 12. It can be seen from Figure 12 that the estimated errors of the grid frequency and the phase differences between phase B&C and phase A voltages are almost zeros. The detection time for the FCs, DCs, and the harmonic voltage amplitudes is less than 40 ms with little overshoots.

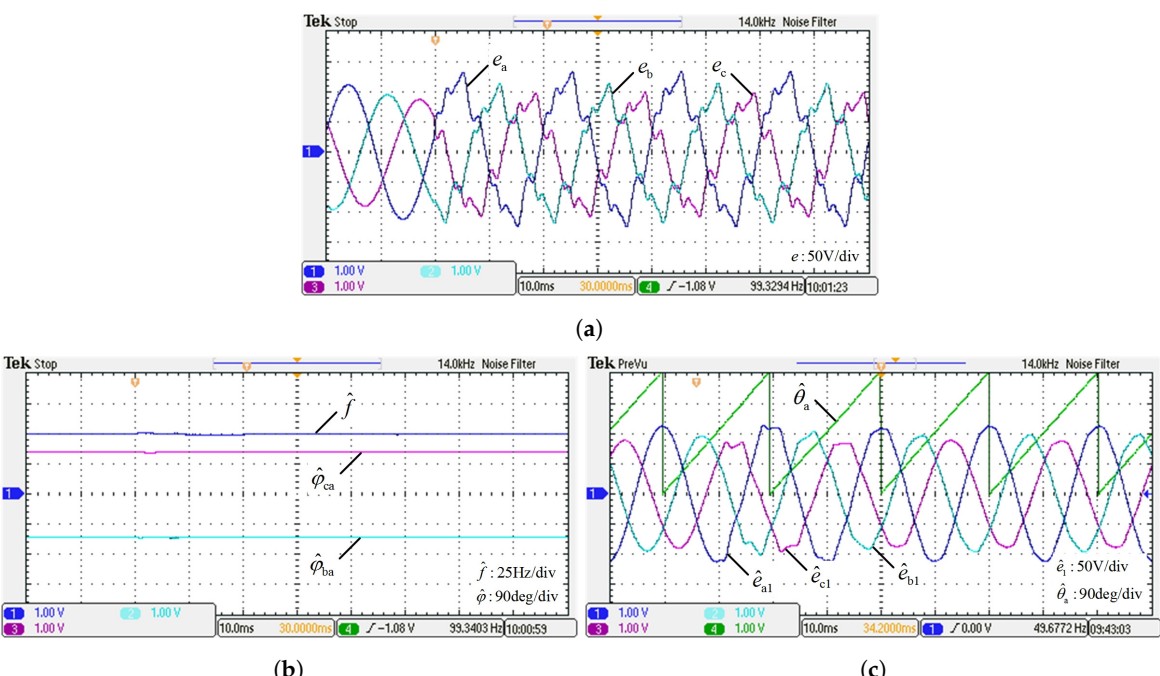

(**a**)

(**b**)             (**c**)

**Figure 12.** *Cont.*

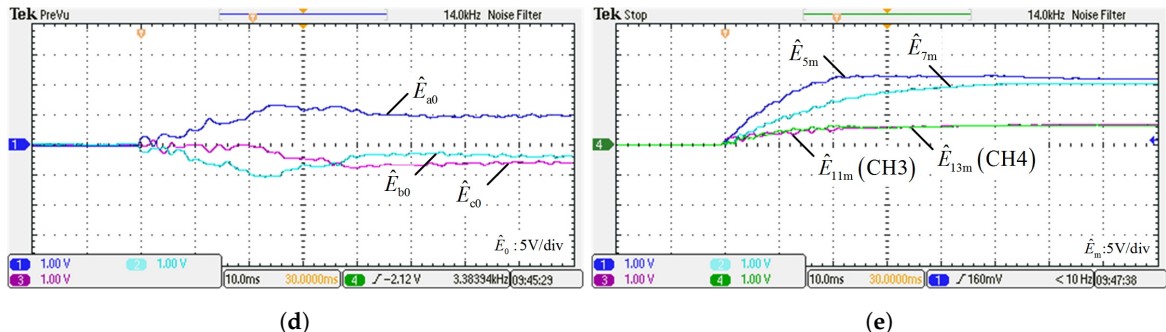

(**d**)                                                   (**e**)

**Figure 12.** Experimental results when three-phase unbalanced voltages are suddenly injected with DC offsets and higher harmonics. The waveforms of (**a**) $e_{abc}$; (**b**) $\hat{f}$, $\hat{\varphi}_{ba}$, and $\hat{\varphi}_{ca}$; (**c**) $\hat{e}_{abc}$ and $\hat{\theta}_a$; (**d**) $\hat{E}_{abc0}$; (**e**) $\hat{E}_{5m}$, $\hat{E}_{7m}$, $\hat{E}_{11m}$, and $\hat{E}_{13m}$.

### 7.2.2. Condition II

When the phase difference between phase B and phase A voltages has a step change from $-130°$ to $-150°$, the experimental results are shown in Figure 13. It can be seen from Figure 13 that the estimated errors of the grid frequency and the phase differences between phase B&C and phase A voltages are very little. The detection time for the FCs, DCs, and the harmonic voltage amplitudes is less than 25 ms, while the estimated DC offsets have large fluctuations and the estimated higher harmonic voltages have little fluctuations.

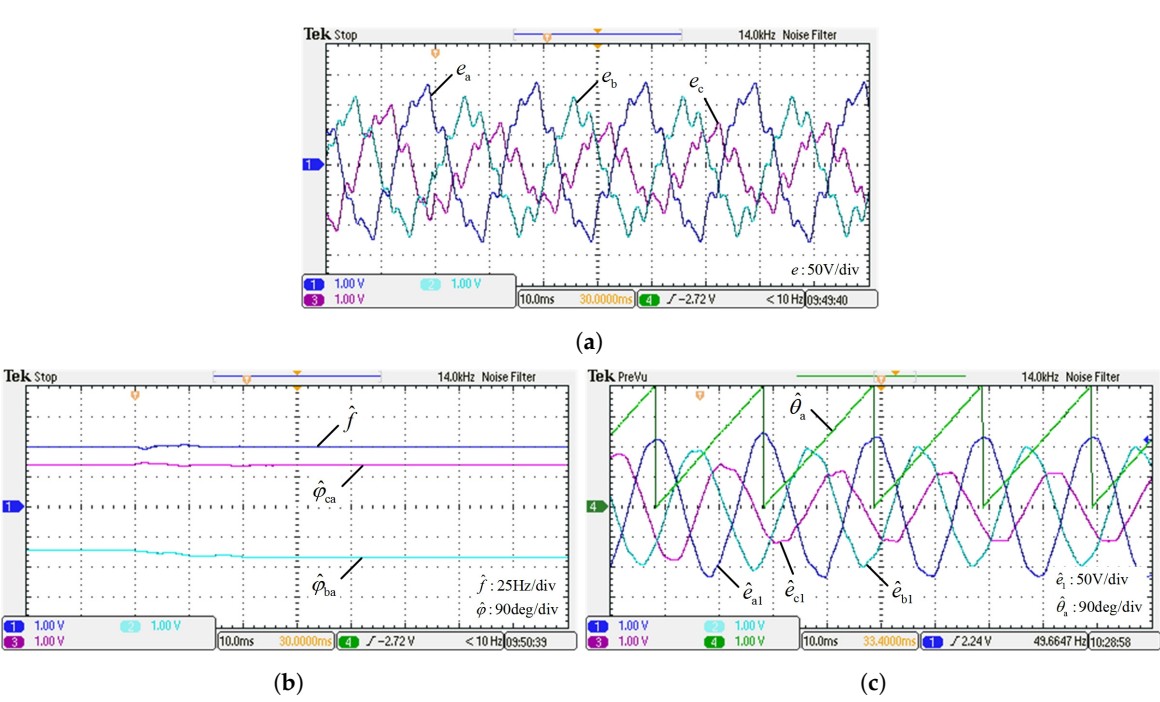

(**a**)

(**b**)                                                   (**c**)

**Figure 13.** *Cont.*

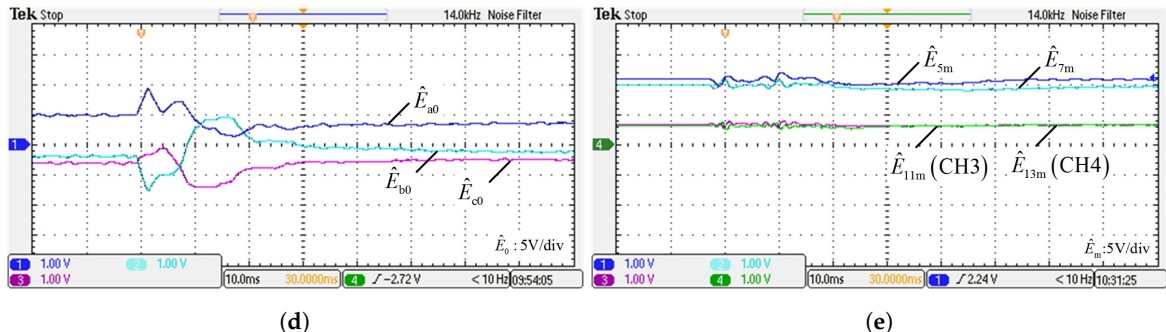

(**d**)  (**e**)

**Figure 13.** Experimental results when $\varphi_{ba}$ has a step change from $-130°$ to $-150°$. The waveforms of (**a**) $e_{abc}$; (**b**) $\hat{f}$, $\hat{\varphi}_{ba}$, and $\hat{\varphi}_{ca}$; (**c**) $\hat{e}_{abc}$ and $\hat{\theta}_a$; (**d**) $\hat{E}_{abc0}$; (**e**) $\hat{E}_{5m}$, $\hat{E}_{7m}$, $\hat{E}_{11m}$, and $\hat{E}_{13m}$.

7.2.3. Condition III

When the grid frequency $f$ has a step change from 50 Hz to 45 Hz, the experimental results are shown in Figure 14. It can be seen from Figure 14 that the estimated errors of the grid frequency and the phase differences between phase B&C and phase A voltages are very little. The detection time for the FCs, DCs, and the harmonic voltage amplitudes is less than 60 ms, while the estimated DC offsets have large fluctuations and the estimated higher harmonic voltages have relatively less fluctuations.

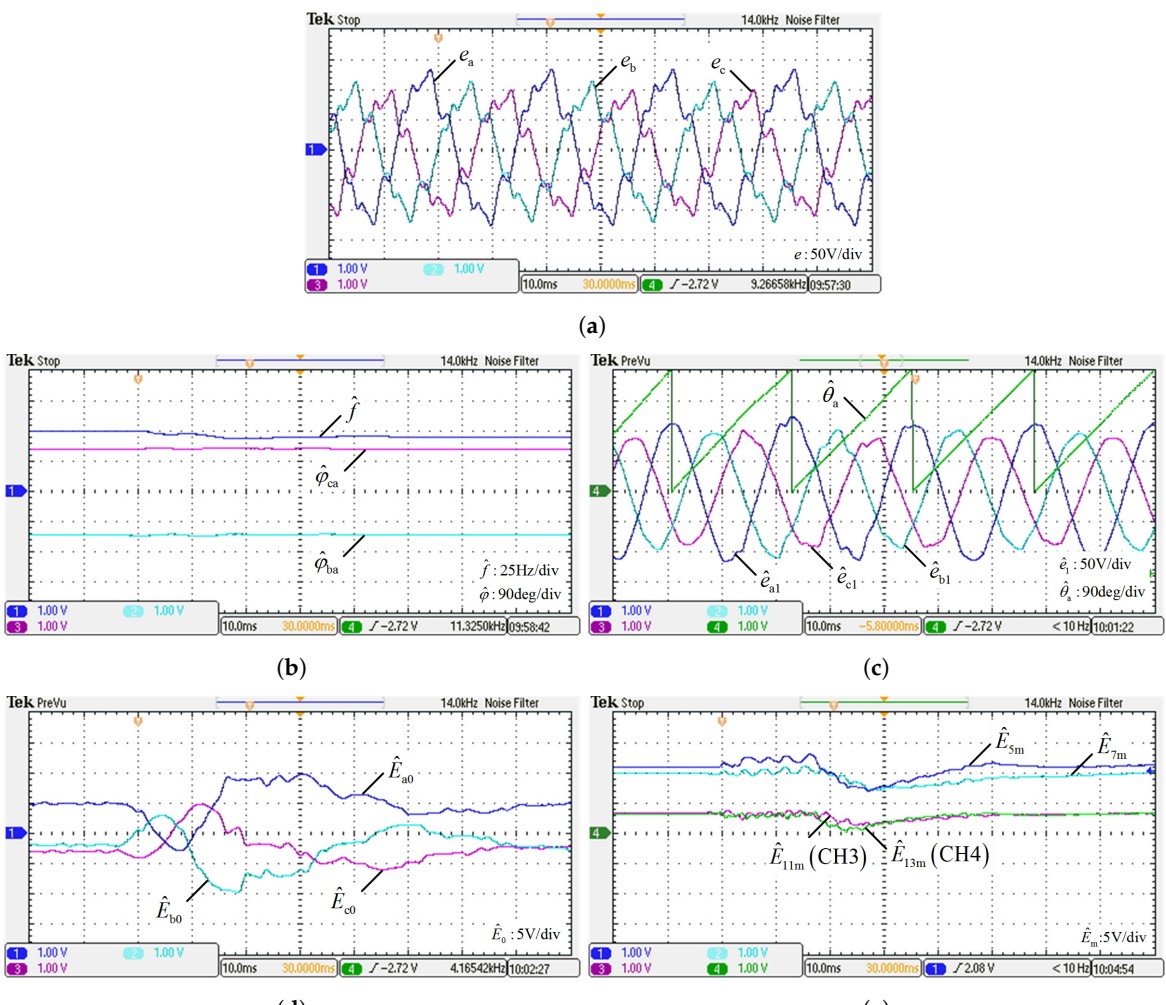

**Figure 14.** Experimental results when the grid frequency $f$ has a step change from 50 Hz to 45 Hz. The waveforms of (**a**) $e_{abc}$; (**b**) $\hat{f}$, $\hat{\varphi}_{ba}$, and $\hat{\varphi}_{ca}$; (**c**) $\hat{e}_{abc}$ and $\hat{\theta}_a$; (**d**) $\hat{E}_{abc0}$; (**e**) $\hat{E}_{5m}$, $\hat{E}_{7m}$, $\hat{E}_{11m}$, and $\hat{E}_{13m}$.

It can be concluded according to the experimental results that the proposed whole PLL system can successfully separate the FCs, DCs, and HCs of the three-phase non-ideal voltages, respectively, and detect all the parameters in less than 60ms under various types of transient conditions. Except for the large fluctuations of the estimated errors in detecting the DC offsets of the three-phase voltages, the overshoots or fluctuations in detecting the other parameters are little. Therefore, the experimental results can demonstrate that the proposed whole PLL system has good steady and transient performance.

## 8. Conclusions

This paper has proposed a PLL technique based on the combination of FC&DC-DS and HC-DS with decoupling network, which can realize the rapid detection of the parameters of FCs, DCs, and HCs of the three-phase non-ideal voltages, respectively. The control law for each estimated parameter can be designed as the first-order or second-order system (grid frequency and instantaneous phase of phase A voltage are the latter case) which is related to only the corresponding estimated parameter error, so the decoupling control is realized for each estimated parameter. The response speed of the whole PLL system can be further improved by setting the control law for each estimated parameter with the same settling time. The proposed PLL technique can directly detect the instantaneous phases of three-phase fundamental voltages other than those of the three-phase PS voltages. It can be concluded from the simulation and experimental results that the proposed PLL technique has good steady performance, and the detection time for each parameter does not exceed three grid cycles in various types of transient conditions, which can demonstrate that the PLL technique also has good transient performance and high stability. Future work will focus on the practical application of the proposed PLL technique to the three-phase AC microgrid.

**Author Contributions:** Conceptualization, G.T. and X.S.; Data curation, G.T. and C.Z.; Formal analysis, G.T. and C.Z.; Funding acquisition, G.T. and X.S.; Investigation, G.T. and C.Z.; Methodology, G.T. and X.S.; Project administration, G.T. and X.S.; Resources, G.T. and X.S.; Software, G.T. and C.Z.; Supervision, G.T.; Validation, G.T. and C.Z.; Visualization, G.T. and C.Z.; Writing—original draft, C.Z.; Writing—review & editing, G.T. and X.S. All authors have read and agreed to the published version of the manuscript.

**Funding:** This research was funded by the National Natural Science Foundation of China grant number 61903320 and 51677162, and the Natural Science Foundation of Hebei Province grant number E2018203174 and E2017203337.

**Conflicts of Interest:** The authors declare no conflict of interest.

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
