# Peer review of "Tan-Sun Transformation-Based Phase-Locked Loop in Detection of the Grid Synchronous Signals under Distorted Grid Conditions"

_electronics, doi:10.3390/electronics9040674_

Round 1

Reviewer 1 Report

Paper deals with the three-phase voltages which are disturbed by unbalance, offsets, etc. Paper introduces two modules decentralized which help in detecting lower and higher harmonics. Before proceeding further with this paper I would like the following comments to be addressed:

  • How are simulation parameters (three-phase nonideal voltages) chosen in Table1.?
  • I am not sure if section 6.2 is a typo or what? The title should be defined differently.
  • Figure 8. It would be interesting to see the behavior when a different variation of phase is present, not only step from 130 to 140 degrees phase
  • The same comment for figure 9. I would like to see more variation in the phase.
  • Figures 11,12,13,14 are tiny/hard to see, I would suggest making them bigger, especially the transient ones.
  • Line 336

Reviewer 2 Report

Dear authors,
thank you for a very comprehensive work. The work has an appropriate level. While it was difficult to follow the text in places, everything is clear and clean when read repeatedly.
I recommend to verify the conclusions of your article by practical deployment in some SMPS application. However, I am happy to recommend the article for publication in the submitted form.

Author Response

Response to Reviewer 2 Comments

Point 1: I recommend to verify the conclusions of your article by practical deployment in some SMPS application. However, I am happy to recommend the article for publication in the submitted form.

Response 1: Thanks for your approval. We will focus on the practical application of the proposed PLL technique to the three-phase AC microgrid in the future work.